# Perception and Willingness to Maintain Continuity of Care by Parents of Children with Asthma in Taiwan

**DOI:** 10.3390/ijerph18073600

**Published:** 2021-03-30

**Authors:** Christy Pu, Yu-Chen Tseng, Gau-Jun Tang, Yen-Hsiung Lin, Chien-Heng Lin, I-Jen Wang

**Affiliations:** 1Institute of Public Health, National Yang Ming Yang Ming Chiao Tung University, Taipei 112304, Taiwan; cypu@ym.edu.tw; 2Institute of Hospital and Health Care Administration, National Yang Ming Chiao Tung University, Taipei 112304, Taiwan; kid_king0422@hotmail.com (Y.-C.T.); gjtang@ym.edu.tw (G.-J.T.); 3Hengchun Tourism Hospital, Ministry of Health and Welfare, Hengchun 946, Taiwan; yhs_lin@yahoo.com.tw; 4Division of Pediatric Pulmonology, Children’s Hospital, China Medical University, Taichung 404, Taiwan; lch227@ms39.hinet.net; 5Department of Pediatrics, Taipei Hospital, Ministry of Health and Welfare, Taipei 242033, Taiwan; 6School of Medicine, National Yang Ming Chiao Tung University, Taipei 112304, Taiwan; 7College of Public Health, China Medical University, Taichung 40402, Taiwan; 8National Institute of Environmental Health Sciences, National Health Research Institutes, Miaoli 350401, Taiwan; 9National Taiwan University, Taipei 10617, Taiwan

**Keywords:** asthma, pediatric care, continuity of care, willingness to pay, national health insurance

## Abstract

To investigate caregivers’ attitudes toward continuity of care (COC) and their willingness to maintain continuity for their children with asthma under a national health insurance (NHI) system without strict referral management. We sampled 825 individuals from six pediatric outpatient departments in different parts of Taiwan from 2017 to 2018. We used a contingent valuation with a payment card method. Post-stratification weighting adjustment and coarsened exact matching were utilized. Multiple logistic regression was used to compare the willingness to pay and spend extra time maintaining continuity by parents. More than 80% of caregivers in the asthma group believed having a primary pediatrician was important for children’s health. Only 27.5% and 15.8% of caregivers in the asthma and control groups, respectively, believed changing pediatricians would negatively affect therapeutic outcomes. Regression analysis showed that the predicted willingness to pay for the asthma and non-asthma groups were NT$508 (SD = 196) and NT$402 (SD = 172), respectively, and there was a significant positive dose–response relationship between household income and willingness to pay for maintaining health care provider continuity. Caregivers’ free choices among health care providers may reduce willingness to spend extra effort to maintain high COC. Caregivers should be educated on the importance of COC.

## 1. Introduction

Asthma is the most common chronic disease among children worldwide. The prevalence rate of asthma has been reported to be as high as 19.7% in children aged 6–7 years, with an average admission rate among asthmatic children of 105.0 per 100,000 population [1]. Asthma can become a severe chronic disease if not well controlled, and poorly controlled asthma increases the risks of emergency department (ED) visits, hospitalization, and school absence, thereby imposing a substantial burden on the patients, their families, and health care systems [2].

Strategies for asthma control include reducing environmental risk factors and a persistent, stepwise approach to pharmacotherapy [3]. Both of these components require coordination between patients and their health care providers. According to the Canadian and International Asthma guidelines, continuity of care (COC) is a key component of asthma management [4]. Provider COC reflects the extent to which patients change physicians, and consulting the same provider is considered to be essential for better patient outcomes because of an improved physician–patient relationship [5]. A previous study reported that apart from low premiums, being able to retain and visit the same health care provider was the most important component when choosing a health plan for consumers [6]. Children with a low COC have been reported to have a higher risk of asthma-specific ED visits [7], and continuity of primary care has been associated with a decrease in asthma hospitalizations in pediatric patients [8]. Moreover, high quality continuity has been associated with better adherence to medications and treatment plans [9,10,11], and seeing the same provider on a continuing basis has been associated with better patient outcomes, such as fewer hospital admissions and reduced medical costs [12,13]. Furthermore, continuity has also been shown to be particularly beneficial for older patients and those with a low income or chronic diseases [5,14,15]. High care continuity has also been shown to improve patient communication and shared decision-making with regards to disease management [16,17].

Even though the importance of COC has been demonstrated, few studies have investigated how parents of children with asthma value COC under a health care system without referral management. Taiwan implemented the National Health Insurance (NHI) system in 1995. The NHI system aims to provide universal health coverage and remove financial barriers for seeking medical care. With a higher copayment, patients can select physicians and specialists freely without referral. Visiting a medical center directly without referral costs approximately US$14; with referral, it costs US$5.6. For clinics, the copayment is US$1.7 per visit (the annual per capita disposable income in Taiwan was approximately US$11,300 in 2018). Copayments are the same with or without referral, because clinics are the lowest level of medical institution in Taiwan. This is not a strict referral system, because patients can choose any physician at any level as long as they are willing to pay for differences in copayment set under the NHI.

Few studies have focused on parents’ perception regarding COC for pediatric patients with asthma. Therefore, the aim of this study was to investigate the main caregivers’ attitudes toward COC for children with asthma, and the association between parents’ perceptions of continuity and their willingness to maintain COC for their children with asthma under a non-strict referral system.

## 2. Materials and Methods

### 2.1. Study Design and Sample Collection

We conducted this cross-sectional study using a purposive sampling design from August 2017 to February 2018. Six medical sites in the northern and central parts of Taiwan were included: two medical centers, two regional hospitals, one district hospital, and one clinic. Medical centers provide the most advanced specialist outpatient services for more serious patients compared to regional hospitals, district hospitals, and clinics. Clinics provide primary health care services to asthma patients with mild and stable conditions. The six sites were chosen to represent different levels of medical institutions with a reasonably diverse geographic distribution. The study participants were pediatric patients’ parents or main caregivers. During the study period, the caregivers and parents who visited the designated study sites were approached, and if they agreed to participate in the study, a self-administered questionnaire was completed on site. Written informed consent was obtained from all respondents. In compliance with the principles of the Helsinki Declaration, the hospital’s Institutional Review Board ratified the study protocol (project identification code: TH-IRB-0017-0013). A trained staff member was available at each study site to answer any questions from the interviewees. Participants were included if they were aged 20 years or older. Because the main focus of this study was on pediatric asthma patients and their parents or main caregivers, individuals with asthma were purposely oversampled. In total, 857 respondents completed questionnaires at the designated medical sites. Thirty-two respondents who indicated that they were neither the parents nor main caregivers of the child were excluded from the study. In addition, because COC is applicable predominantly to disease outcomes, we excluded the respondents whose children had visited physicians only as outpatients for regular growth-evaluation purposes (*n* = 74) to conduct regression analysis. After coarsened exact matching (explained below), 146 respondents were included in the asthma group and 547 in the non-asthma group in regression analysis.

### 2.2. Survey Measures and Instrument

The questionnaire was designed by the researcher and validated by six experts who were either pediatric physicians or professors in public health. The questionnaire was pretested on 30 respondents and modified according to pretest results. The survey comprised a series of closed-ended questions, and the asthma status of the children was self-reported by the respondent. The children were confirmed to have asthma if they were diagnosed with asthma by a physician and were prescribed with asthmatic pharmacotherapy. Both criteria had to be met for a child to be defined as having asthma.

Perceptions and values of continuity in the parents and caregivers were measured using seven items. The level of provider continuity was measured through a self-reported question on the frequency of changing pediatric physicians for the same disease: “During the past year, how often have you changed pediatric physician for the same disease?” To analyze the caregivers’ perceptions of interpersonal trust toward the doctors and the importance of continuity, the following questions were asked: “Whether or not you have changed pediatric physician, do you think changing physician would affect the outcome of your child’s therapy?” (answers: “Yes, it would have a negative effect”, “No, it would not affect the therapeutic outcome”, or “Yes, it would have a positive effect”); “Score your confidence level in your pediatric physician during the past year.” (answers on a 5-point Likert scale: 1 = no trust, 5 = very trusting); and “Do you think having a primary pediatric physician for your child is important?” (answers: “unimportant”, “somewhat important”, or “very important”). We then asked the respondents about their reasons for changing pediatricians during the past year. This was a multiple choice question that asked the respondents to tick the three most applicable reasons.

We used the contingent valuation method [18,19] to evaluate willingness to incur expense and willingness to spend extra time maintaining COC for the respondents’ children. Willingness to spend time maintaining provider continuity was assessed using the question, “If the physician your child usually visits relocated, such that it would take longer to reach him or her, would you spend more than 30 min travelling for your child to see this physician?” In a separate question, the respondents were asked to elicit their maximum willingness to pay (WTP) for maintaining continuity with the same physician using a payment card (NT$0, NT$1–100, NT$101–300, NT$301–600, NT$601–1000, and more than NT$1000). To estimate WTP, we estimated a linear model with the control variables (detailed in Table 1) and used the higher value in the range of the WTP categories as the dependent variable. 

### 2.3. Statistical Analysis

We used post-stratification weighting to calculate the population estimates. Post-stratification is a common technique used in survey analysis for accurately estimating population distribution [20]. To adjust the unequal probabilities of selection into the sample for population units of analysis [21], our study samples were weighted with post-stratification adjustments according to the children’s age and sex based on the 2016 annual report released by the Ministry of Health and Welfare in Taiwan. Considering that the missing percentage was low (less than 5% in each person) [22], we applied listwise deletion during all inferential analysis.

We used coarsened exact matching (CEM) [23] to account for differences in characteristics between the case and control groups. In CEM, we first coarsened our data temporarily and then matched our coarsened data using exact matching. The matching variables were the caregivers’ sex, age, education, household income, child’s age, and child’s sex. The matched samples included in the regression analysis after matching were 146 in the asthma group and 547 in the non-asthma group (Table 1). We then examined the association between willingness to pay and a parent/main caregiver’s perception regarding continuity using multiple ordered logistic regression models stratified by the children’s asthma status. The original four categories were retained for willingness to pay to avoid loss of information. All analyses were conducted using STATA 15.0 software (College Station, TX, USA). A two-sided α-level of <0.05 was considered to be statistically significant.

## 3. Results

Table 1 shows the characteristics of the participants before and after CEM matching by asthma groups.

Table 2 shows the results for willingness to spend extra money and time to be able to see the same physician in cases where the original physician relocates. A higher proportion of parents/main caregivers in the asthma group reported that they would be willing to spend an extra 30 min to visit their original physician compared to the control group (54.8% vs. 44.2%, *p* = 0.021). The distribution of the willingness to pay to see the same physician appeared to be skewed more toward higher values for the asthma group (*p* = 0.001).

Table 2 also shows that 27.5% and 15.8% of the respondents in the asthma and non-asthma groups, respectively, indicated that they thought changing pediatricians would have a negative effect on therapeutic outcomes for their children. A slightly higher proportion of respondents in the asthma group indicated that they thought changing pediatricians would have a positive effect on therapeutic outcome (30.9% for the asthma group and 29.4% for the control group).

Ordinary least squares for willingness to pay by asthma status is reported in Table 3. The predicted willingness to pay for the asthma and non-asthma groups were NT$508 (SD = 196) and NT$402 (SD = 172), respectively. The parents/main caregivers’ perceptions regarding whether changing pediatricians would affect therapeutic outcomes were not significantly associated with willingness to pay in both groups. In both groups, respondents who valued COC had higher willingness to pay to maintain care continuity for their children. Trust in a pediatrician was significantly associated with willingness to pay only in the control group. Male parents or main caregivers were significantly more likely to pay a higher amount to maintain care continuity in the asthma group. Household income was significantly associated with willingness to pay in both groups and with a dose-response effect, where higher willingness to pay was observed for higher income households. 

Figure 1a,b shows the predicted willingness to pay by monthly household income. There was a clear dose-response effect of income on willingness to pay. However, the opposite was found for willingness to spend extra time (Table 4). In the asthma group, respondents from higher income households were less willing to spend extra time to maintain COC with an existing physician than respondents from lower income households; this was not observed in the control group.

The level of medical institution was significantly related to willingness to spend extra money and time. Individuals sampled from the district hospital or clinic were more willing to spend extra money to maintain COC (OR = 7.35, *p* < 0.01); however, this group was less willing to spend extra time.

## 4. Discussion

COC is not only a fundamental characteristic of high-quality care but also a key component of disease management in asthma. Patient–physician relationships are crucial and have a positive effect on health outcomes, and provider continuity has been shown to improve patient outcomes when trust and confidence in pediatricians have been established [7,8]. To the best of our knowledge, this is the first survey of parents and caregivers’ attitudes toward provider continuity and their willingness to spend time and money to maintain COC for children with asthma.

An important aspect of this study is that other countries, such as South Korea and Japan, also operate a non-strict referral system, similar to Taiwan. Under this kind of system, patients can transfer easily to other providers, and this may affect their willingness to maintain high COC with a particular provider, negatively affecting patient outcomes. In addition, we observed that for both the asthma and control groups, household income was a significant determinant of willingness to pay for care continuity. This information is meaningful, because although universal health coverage in Taiwan is provided by the NHI, finances still affect parents’ decisions regarding provider continuity.

Maintaining continuity is particularly important for patients with asthma [8,24,25]. For children, asthma management may be different than for adults, because caregivers are the key link between health care providers and children with asthma [26], and they are often the key decision-makers with regards to medical care. One of our main findings was that although the majority of our respondents stated that they thought a primary pediatric physician was important for their children, only a small percentage of the respondents in the asthma group thought that changing pediatric physician would negatively affect therapeutic outcomes. This implies that the respondents did not think changing physician would negatively affect care continuity, which contradicts the definition of provider continuity. This may be due to the ease of access to alternative providers under the health care system in Taiwan.

If COC is to be promoted to parents, they may have to be educated on the importance of COC. Studies on specific diseases have revealed that providing parents with information or education can have positive effects on children’s health outcomes [27,28]. Our findings support these studies by demonstrating that poor COC for the children may have resulted from low awareness regarding the importance of COC among the parents and caregivers.

We found that the respondents in the asthma group who were recruited from district hospitals and clinics had higher willingness to pay and were less willing to spend time to maintain continuity when compared with the respondents recruited from other medical institutions. In Taiwan, while medical centers provide more sophisticated medical procedures and a wider range of services, it also takes more time to go to medical centers for treatment. Clinics and district hospitals, on the other hand, provide a reasonable quality of care without complicated administrative procedures. Thus, the respondents recruited from clinics and district hospitals may have involved parents/main caregivers who had higher opportunity costs for spending time. We also found that the respondents in the asthma group had slightly higher predicted willingness to pay than the non-asthma group. This is expected, as worse health conditions are more likely to benefit from higher care continuity.

Provider continuity has been positively associated with trust in the physician, belief in the effect of changing physician, and the value placed on COC. When a patient regularly sees the same doctor, they are more willing to share their concerns and expectations with them, and in turn, the physician is more likely to have accumulated knowledge about that patient’s clinical condition and preferences [5]. Patient trust may be an essential component of a parents’ perception of continuity. When parents trust their children’s physician, they are more likely to pay more to see the same physician. A clear causal relationship has been shown between a patient’s trust and provider continuity [16]. We also found that many respondents believed that changing physicians could lead to better outcomes. This may be because medical care in Taiwan is highly specialized, and thus patients may prefer to visit different specialists for different health conditions.

Other studies have investigated WTP by parents for obtaining integrated care or access to care for their children. For example, Poder and He [29] examined WTP by parents for interdisciplinary musculoskeletal clinics in Canada. They found that the study population had a mean WTP of 42.3 Canadian dollars per person for such clinics, and this corresponded to a mean reduction from 12 to 4 months of wait time. Regier et al. [30] also found that parents had significantly higher WTP when waiting time for obtaining diagnosis was reduced.

There are several limitations to this study. First, we collected our sample from the northern and central parts of Taiwan, and thus the results may not be generalizable to the whole of Taiwan. However, we weighted our data using an outpatient database of the whole population, which improved the representativeness of our sample. Second, this is a cross-sectional survey, and hence, longitudinal changes cannot be observed. Third, response and recall biases were possible. However, this is a typical problem in survey studies. Finally, we did not collect information on children’s diseases other than asthma, and other health conditions may have affected the comparability of the two study groups. However, we believe this should not be a major problem in our sample because the respondents were recruited from general outpatients. In Taiwan, medical care is highly specialized, and thus, if a child has a major disease such as cancer or heart disease, they are likely to be treated in specialized outpatient settings.

Compared with general participants, the parents who had children with asthma were not willing to spend more time or money. Our study did not only investigate patients’ WTP for COC but also analyzed the relationship between the parents’ perception regarding and value attributed to continuity. The parents and main caregivers did not think that changing physician would affect care continuity. Future studies should continue to explore the causal linkage between continuity and value that parents place on continuity. Furthermore, researchers can investigate physicians’ perception of COC and compare providers’ characteristics, which may also be important determinants of parents’ attitudes toward their children’s COC.

## 5. Conclusions

In conclusion, we found that the respondents in the asthma group had a slightly higher predicted willingness to pay than the non-asthma group. Changing a doctor may affect the relationship between the doctor and patient and increase medical costs. Therefore, the government should implement dedicated physician programs for children. Pediatricians can serve as special physicians for children and be actively involved in evaluating their development, vaccinations, and other health indicators. Since the use of pediatric medical care mostly depends on changes in demand, a flexible policy should be implemented, and caregivers should be educated. We recommend that the government advocate the importance of COC. Moreover, we also recommend promoting hierarchical medical care, which can help to integrate medical care. Taken together, these steps can provide children with comprehensive care and better COC.

## Figures and Tables

**Figure 1 ijerph-18-03600-f001:**
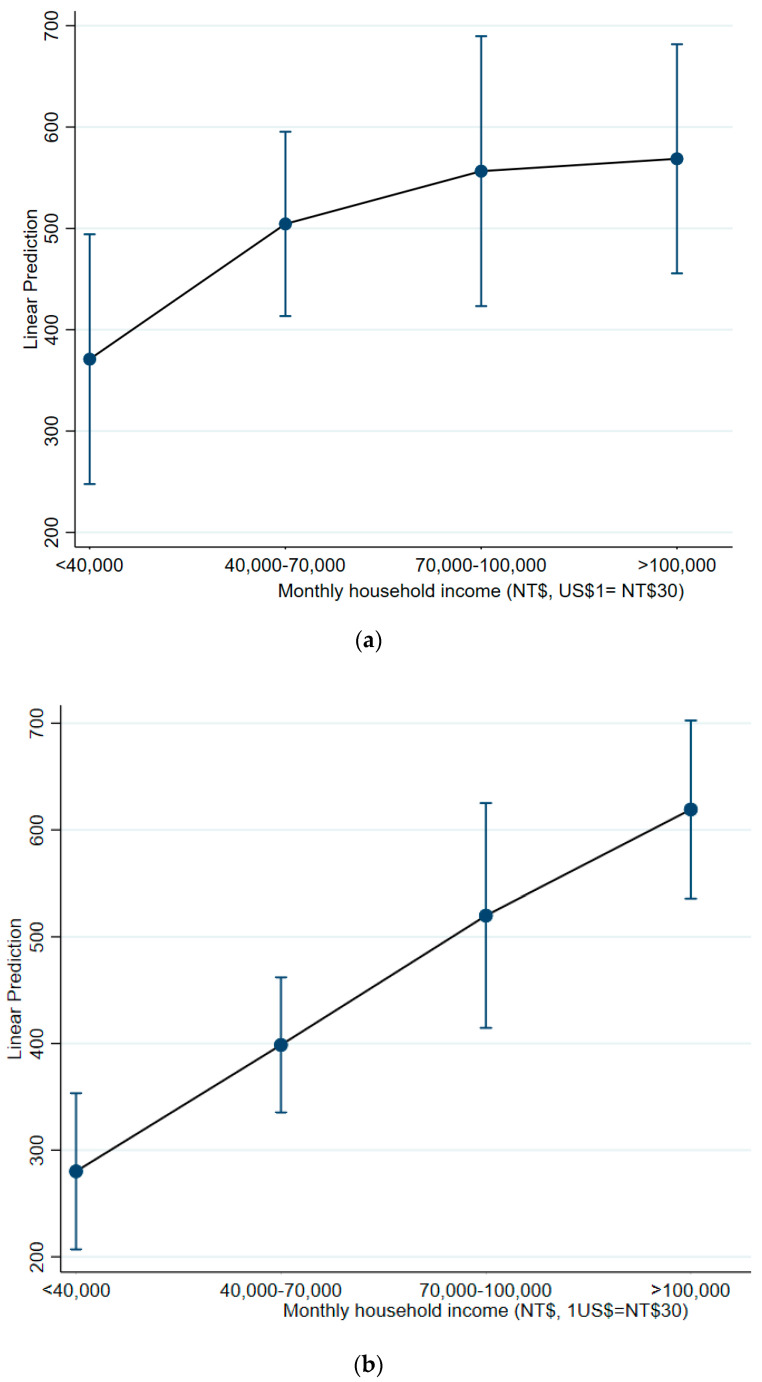
(**a**) Predicted willingness to pay by income, asthma group; (**b**) Predicted willingness to pay by income, non-asthma group.

**Table 1 ijerph-18-03600-t001:** Participants’ characteristics matched by asthma status in children.

	Pre-Coursened Exact Matching (CEM) (*n* = 751)			Post-CEM (*n* = 693)		
	With Asthma	Without Asthma	*p* Value	With Asthma	Without Asthma	*p* Value
Sample size	*n* = 147	*n* = 604		*n* = 146	*n* = 547	
Caregiver/parent						
Parent	143(98.5)	555(94.0)	0.004	142(98.5)	502(95.1)	0.022
Age (y); mean (SD)	43(0.7)	42(0.4)	0.089	43(0.7)	42(0.4)	0.290
Female sex	116(73.4)	472(79.8)	0.297	117(73.6)	508(84.1)	0.064
College degree	100(61.8)	351(50.0)	0.101	99(61.9)	324(62.6)	0.921
Monthly household income			0.145			
<40,000	32(22.6)	158(28.4)		32(22.6)	145(21.8)	
40,000–70,000	45(35.4)	236(36.7)		45(35.5)	223(31.5)	
70,000–100,000	37(18.1)	125(21.8)		36(18.0)	103(27.0)	
>100,000	33(23.9)	80(13.1)		33(23.9)	76(19.7)	
Have married	134(91.0)	562(91.2)	0.883	133(91.1)	511(94.3)	0.387
Children						
Age (y); mean (SD)	11(0.6)	9(0.2)	0.031	11(0.6)	10(0.4)	0.304
Female sex	52(42.7)	264(49.8)	0.321	51(42.5)	235(40.4)	0.775
With severe illness	4(4.7)	29(5.0)	0.940	4(4.7)	27(5.5)	0.842
Caregiver rated child’s health status			0.375			
Good	27(21.0)	205(29.0)		26(20.5)	177(26.6)	
Fair	87(60.5)	321(56.9)		87(60.8)	296(57.0)	
Poor	33(18.5)	75(14.1)		33(18.7)	71(16.4)	
Sites of sample collection			<0.001			<0.001
Medical center	58(38.5)	73(15.9)		58(38.7)	65(18.4)	
Regional	71(48.0)	360(63.7)		70(47.8)	328(66.2)	
District and clinic	18(13.5)	171(20.4)		18(13.5)	154(15.4)	

Data are given as numbers (weighting percentages) of patients unless otherwise indicated.

**Table 2 ijerph-18-03600-t002:** Willingness to spend additional money and time to avoid changing physicians.

		With Asthma (*n* = 146) %	Without Asthma (*n* = 547) %	*p* Value ^1^
**Parent’s perception of continuity (weighted)**				
Willingness to pay money				
0		5(3.4)	36(6.6)	0.001
1–300		67(45.9)	328(60.0)	
301–600		35(24.0)	94(17.2)	
>600		38(26.0)	86(15.7)	
No response		1(0.7)	3(0.5)	
Willingness to spend time				
<30 min		65 (44.5)	303 (55.4)	0.021
>30 min		80 (54.8)	242 (44.2)	
No responses		1(0.7)	2(0.4)	
**Parent’s perception of continuity (unweighted)**				
1. During the past year, how often have you changed pediatric physicians for the same disease?	Never	55(36.9)	158(23.4)	<0.001
	Seldom	83(55.7)	417(61.7)	
	Often	8(5.4)	24(3.6)	
	Always	1(0.7)	3(0.4)	
	NA	2(1.3)	72(10.7)	
	No response	0(0.0)	2(0.3)	
2. Whether you have changed the pediatric physicians or not, do you think that changing physicians affect your child’s therapeutic outcomes?	It would have a negative effect	41(27.5)	107(15.8)	0.001
	It would not affect outcome	61(40.9)	365(54.0)	
	It would have a positive effect	46(30.9)	199(29.4)	
	No response	1(0.7)	5(0.7)	
3. Do you think having a primary pediatric physician is important for your child?	Unimportant	1(0.7)	8(1.2)	0.279
	Somewhat	15(10.1)	99(14.6)	
	Very important	133(89.3)	567(83.9)	
	No response	0(0.0)	2(0.3)	
4. Score an overall confidence level for all the pediatric physicians you have met during the past year	1	1(0.7)	2(0.3)	0.051
	2	4(2.7)	11(1.6)	
	3	23(15.4)	156(23.1)	
	4	65(43.6)	322(47.6)	
	5	56(37.6)	184(27.2)	
	No response	0(0.0)	1(0.1)	

^1^ Excluded no-responses.

**Table 3 ijerph-18-03600-t003:** Linear regression for willingness to pay.

	With Asthma (*n* = 144)		Without Asthma (*n* = 534)	
Characteristics	Estimates	95% CI	Estimates	95% CI
Provider continuity level (Ref: Never changed)				
Have changed	−5.65	[−136.90,125.61]	−21.59	[−108.44,65.26]
Parent’s perception of continuity				
Effect of changing physicians for therapy (Ref: Positive)				
Negative	72.89	[−136.90,125.61]	80.88	[−31.93,193.69]
No effect	23.41	[−101.47,148.29]	−50.79	[−145.67,44.09]
Continuity is important (Ref: Unimportant)	199.28 ***	[83.14,315.42]	145.53 **	[49.67,241.39]
Trust in physician (Ref: Non-trust)	141.00	[−16.51,298.50]	116.70 ***	[48.33,185.07]
Parents/caregivers				
Male sex (Ref: female)	254.14 ***	[118.85,389.44]	53.07	[−33.93,140.08]
Age	−3.44	[−12.81,5.92]	−1.19	[−5.37,3.00]
College degree (Ref: high school)	−76.29	[−218.43,65.85]	−17.41	[−115.48,80.66]
Monthly household income (Ref: <NTD40,000)				
40,000–70,000	133.46	[−15.09,282.01]	118.49 **	[28.93,208.04]
70,000–100,000	185.53	[−7.02,378.09]	239.75 ***	[101.78,377.72]
>100,000	197.73 *	[7.76,387.69]	339.02 ***	[221.19,456.86]
Children				
Male sex (Ref: female)	13.50	[−92.05,119.05]	−10.81	[−91.35,69.73]
Age > 6 y	−1.66	[−15.79,12.47]	12.88 **	[4.56,21.19]
Caregiver-reported child’s health (Ref: Good)				
Poor	68.09	[−128.70,264.88]	35.45	[−96.47,167.36]
Fair	61.65	[−62.99,186.29]	−57.76	[−151.03,35.51]
Site of sample collection (Ref: Medical center)				
Regional	3.05	[−129.83,135.92]	15.04	[−91.61,121.69]
District and clinic	242.58 *	[57.88,427.28]	132.80 *	[18.32,247.28]

* *p* <0.5, ** *p* < 0.01, *** *p* < 0.001.

**Table 4 ijerph-18-03600-t004:** Willingness to spend time for maintaining continuity according to asthma status.

	With Asthma (*n* = 143)		Without Asthma (*n* = 533)	
Characteristics	Odds Ratio	95% CI	Odds Ratio	95% CI
Provider continuity level (Ref: Never changed)				
Have changed	1.25	[0.49,3.23]	0.58	[0.31,1.11]
Parent’s perception of continuity				
Effect of changing physicians for therapy (Ref: Positive)				
Negative	1.26	[0.34,4.64]	2.31 *	[1.00,5.33]
No effect	1.47	[0.52,4.14]	1.05	[0.56,1.95]
Continuity is important (Ref: Unimportant)	1.70	[0.48,6.02]	0.67	[0.30,1.49]
Trust in physician (Ref: Non-trust)	3.09	[0.88,10.89]	2.37 *	[1.22,4.58]
Parents/caregivers				
Male sex (Ref: female)	3.95 *	[1.30,12.01]	1.26	[0.69,2.31]
Age > 40 y (Ref: ≤39 years)	1.02	[0.95,1.10]	1.01	[0.97,1.06]
College degree (Ref: high school)	0.51	[0.17,1.53]	0.31 ***	[0.16,0.60]
Monthly household income (Ref: <NTD40,000)				
40,000–70,000	0.81	[0.23,2.87]	1.04	[0.48,2.25]
70,000–100,000	0.12 **	[0.02,0.56]	1.00	[0.37,2.67]
>100,000	0.24	[0.05,1.07]	2.35	[0.91,6.05]
Children				
Male sex (Ref: female)	1.81	[0.74,4.42]	0.96	[0.54,1.69]
Age > 6 y (Ref: ≤5 y)	0.88 *	[0.79,0.97]	0.94 *	[0.88,0.99]
Caregiver-reported child’s health (Ref: Good)				
Poor	0.46	[0.09,2.29]	4.81 ***	[2.01,11.48]
Fair	0.37	[0.12,1.11]	1.07	[0.57,2.00]
Site of sample collection (Ref: Medical center)				
Regional	1.64	[0.34,8.03]	0.82	[0.35,1.92]
District and clinic	0.12 **	[0.03,0.53]	0.48 *	[0.26,0.89]

* *p* < 0.5, ** *p* < 0.01, *** *p* < 0.001.

## Data Availability

Data used in this study are not publically available due to IRB regulations.

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
