# Peer review of "Perception and Willingness to Maintain Continuity of Care by Parents of Children with Asthma in Taiwan"

_ijerph, 2021, doi:10.3390/ijerph18073600_

Round 1

Reviewer 1 Report

Considering that you asked the maximum amount, you should use the higher value in the range.

One method to estimate the WTP using econometrics is ordinary least squares (OLS).

WTP = constant + coefficient*control_variable

Since your CV method is almost like a payment card you can also try a logit

WTP(yes/no) = constant + coefficient*bid_price + coefficient*control_variable

Other exemples:

https://www.researchgate.net/publication/257559490_Willingness_to_pay_for_physician_services_Comparing_estimates_from_a_discrete_choice_experiment_and_contingent_valuation

https://bmchealthservres.biomedcentral.com/track/pdf/10.1186/1472-6963-13-208.pdf

Reviewer 2 Report

This manuscript is much improved. I think it's an important topic and the findings are interesting. I don't have any major suggestions for improvement. I think the conclusion is still somewhat unclear, but overall the discussion section is better.

Reviewer 3 Report

None

Reviewer 4 Report

The fluency of the revised manuscript has been improved and I have no further comments.

Round 2

Reviewer 1 Report

Thanks for this revised version.

Note that the Tables should be inserted after they are mentioned in the text (e.g. Table 3). Idem for Figures.

In the methods about WTP estimate, you mentioned linear regression, could you be more specific (OLS)?

In the discussion, please discuss your results of WTP as compared to other studies, even not directly about continuity of care in children, you may find studies about access and continuity of care in other domains (e.g. https://pubmed.ncbi.nlm.nih.gov/27878690/). Also dicuss the hypothetical bias (e.g. https://link.springer.com/article/10.1007/s10640-004-3332-z). Should we apply the divide by 2 rule of the NOAA panel?

Author Response

This manuscript is a resubmission of an earlier submission. The following is a list of the peer review reports and author responses from that submission.

Round 1

Reviewer 1 Report

Interesting manuscript that used CEM.

However, no WTP value is calculated. Econometric methods exist for this.

You only gave attention to the determinants of WTP, but to calculate appropriately the WTP is of utmost importance.

In the abstract, not “countries” but “country”. This is an example of the many mistakes you did in the manuscript. Please revise the English.

In the abstract too, please mention that you used the contingent valuation method, and that it was a referendum format (or a payment card, this is unclear in the Methods).

In the Study design and sample collection Section, please indicate that the six medical sites are in Taiwan. Also, it would be netter to mention this in the title.

Author Response

Dear Editor and Reviewers,
Thank you for the valuable comments on our manuscript. They have indeed helped us enhance our manuscript. 

  • Q1: Interesting manuscript that used CEM. However, no WTP value is calculated. Econometric methods exist for this. You only gave attention to the determinants of WTP, but to calculate appropriately the WTP is of utmost importance.
  • A1: Thank you. The WTP question in this study was obtained by using a payment card method, where the respondents were asked the question “If the physicians your children usually visit relocated, such that it would take longer to reach him or her, what is the maximum amount you would be willing to pay per month for him or her to maintain continuity with the same physician? (NT$0, NT$1–100, NT$101–300, NT$301–600, NT$601–1,000, and more than NT$1,000).” This is a multiple choice question and the respondent can choose from one of the above splines. Their responses were recorded as it is. For example, if a respondent chooses “NT$101-300”, then his/her WTP is coded as “NT$101-300” without further calculation or adjustment. We agree that readers may be confused about the actual method used, and thus we have added the above description in the revised manuscript to ensure clarity.

  • Q2: In the abstract, not “countries” but “country”. This is an example of the many mistakes you did in the manuscript. Please revise the English.
  • A2: Thank you. We have corrected the abstract and other places in the revised manuscript. We have corrected to “country”. Please see lines 17 and 191.

  • Q3: In the abstract too, please mention that you used the contingent valuation method, and that it was a referendum format (or a payment card, this is unclear in the Methods).
  • A3: Thank you. We used a contingent valuation with a payment card method, and we have added this in the abstract and also in Method.

  • Q4: In the Study design and sample collection Section, please indicate that the six medical sites are in Taiwan. Also, it would be netter to mention this in the title.
  • A4: Thank you. This information is added in the revised manuscript as advised. We have mention this in the study design and sample collection section and title. Please see Line 71-75. Six medical sites from the northern and central parts of Taiwan were included: two medical centers, two regional hospitals, one district hospital, and one clinic. Medical centers may provide the most advanced specialist outpatient services to more serious patients than regional hospitals, district hospitals, and clinics. Clinics may provide primary health care services to asthma patients with mild and stable condition. The six sites were chosen to represent different levels of medical institutions with a reasonably diverse geographic distribution.

We have revised the manuscript based on these comments. All changes are tracked in a different color.Please see the attachment.

Thank you again for the valuable comments. We look forward to hear from you.

Reviewer 2 Report

This is a survey study investigating caregivers' attitude toward continuity of care for children with asthma. This is a novel study and I think will have broad interest.

Methods: I am not familiar with the coarsened exact matching technique used by the authors. It might be helpful to provide readers with a line or two of explanation.

Results: It would be more standard to make table 2 table 1 as it summarizes the general demographics of the population. I am unclear what the "Parent" category is in table 2--is that parent vs. caregiver? That could be clarified.

My main critique of this manuscript is that trends are being presented as findings. For example, the authors imply there are differences in responses from the asthma group vs control in their willingness to spend more time and money to see the same physician, but the analysis itself did not reach statistical significance. There are numerous places where this occurs. The authors really cannot demonstrate that there is a difference, and thus it would be better to either state there is not a difference, or to say the percentages and then state that it did not reach statistical significance. There also seems to be a typo in table 2 (P value1).

Table s1 seems important--could that be merged with table 2? I am unable to see it, but it seems too important to be in the appendix. Again, the authors mention percentages but it is unclear whether any of these findings in table s1 are significant?

Discussion: The discussion section only mentions a couple of key findings from the paper, so we are left without a theory about some of the main outcomes from the regression analysis and tables 1 and s1. This section should be expanded and more detail provided. It also seems odd to finish the main discussion paragraph by stating that the main contribution is that parents are unaware of the importance of COC when that was not really the main focus of the paper, or that response even presented in the text. In fact, many caregivers seemed to feel it would be better to change pediatricians--it would be interesting to know why.

Overall this is an interesting and well-written paper. I think the results and discussion should be revised, and the English-language does need some minor editing.

Author Response

Dear Editor and Reviewers,
Thank you for the valuable comments on our manuscript. They have indeed helped us enhance our manuscript.

  • Q1: Not familiar with the coarsened exact matching technique used by the authors. It might be helpful to provide readers with a line or two of explanation. (Methods)
  • A1: Thank you. This information is added in the revised manuscript as advised. Specifically, we added “In CEM, we first coarsened our data temporarily and then we matched our coarsened data using exact matching.”

  • Q2: It would be more standard to make table 2 table 1 as it summarizes the general demographics of the population. (Results)
  • A2: Thank you. As advised, we have changed the order of the tables in the revised manuscript. Please see page 4 and 5.

  • Q3: What the "Parent" category is in table 2--is that parent vs. caregiver? That could be clarified. (Results)
  • A3: Thank you. We meant “parents or main caregivers”. This has been clarified in the revised manuscript.

  • Q4: Critique of this manuscript is that trends are being presented as findings. For example, the authors imply there are differences in responses from the asthma group vs control in their willingness to spend more time and money to see the same physician, but the analysis itself did not reach statistical significance. There are numerous places where this occurs. The authors really cannot demonstrate that there is a difference, and thus it would be better to either state there is not a difference, or to say the percentages and then state that it did not reach statistical significance. There also seems to be a typo in table 2 (P value1).
  • A4: Thank you for pointing this out. We have made some typos in Table 2. We have corrected the numbers in the revised version. The correct numbers indicate there are statistical significance in the distributions between the two groups. We have modified the text in the revised version accordingly.

  • Q5: Table s1 seems important--could that be merged with table 2? I am unable to see it, but it seems too important to be in the appendix. Again, the authors mention percentages but it is unclear whether any of these findings in table s1 are significant?
  • A5: Thank you. As advised, we have merged Table S1 with Table 2 in the revised version. We have also added p-values for all the variables and modified the text accordingly.

  • Q6: The discussion section only mentions a couple of key findings from the paper, so we are left without a theory about some of the main outcomes from the regression analysis and tables 1 and s1. This section should be expanded and more detail provided. It also seems odd to finish the main discussion paragraph by stating that the main contribution is that parents are unaware of the importance of COC when that was not really the main focus of the paper, or that response even presented in the text. In fact, many caregivers seemed to feel it would be better to change pediatricians--it would be interesting to know why. (Discussion)
  • A6: Thank you. We have add more discussion in the revised version. We think the main contribution of our study is on the willingness to pay and spend time to maintain good COC. We thus added some discussion on determinants of willingness in terms of physician trust. Specifically, provider continuity has a positive association with trust in physicians, the belief in the effect of changing physicians, and the value placed on COC. When a patient regularly sees the same doctor, they are more willing to share their concerns and expectations to their medical provider, and in turn, the physician is more likely to establish accumulated knowledge about that patient’s clinical condition and preference. Patients' trust may be one of the essential components in parents’ perception of continuity. When parents trust their children’s physicians, they are more likely to pay extra money to seek the regular pediatric physicians. Even though a patient's trust does show a clear causal relationship with provider continuity, it is generally believed that the importance of trust in a patient–physician relationship is less controversial. We found that many respondents believe changing physician can lead to better outcome. It can be explained by the fact that medical care in Taiwan is highly specialized, thus patients would prefer to visit different specialists than general outpatient.

  • Q7: English-language does need some minor editing.
  • A7: We have edited the English-language of the manuscript.

We have revised the manuscript based on these comments. All changes are tracked in a different color.Please see the attachment.

Thank you again for the valuable comments. We look forward to hear from you.

Reviewer 3 Report

  • Add a limitation section
  • The authors appears to generalize the findings but they are not in a position to do so based on non random selection. 
  • Suggested to add that the findings are limited to the sample tested
  • Extensive english grammar, style and format  changes are required. Highly recommended to have the manuscript reviewed and edited for language by an expert. 
  • Overall, a decent research study  and addresses a very relevant topic faced by the health care systems across the world.

Author Response

Dear Editor and Reviewers,
Thank you for the valuable comments on our manuscript. They have indeed helped us enhance our manuscript.

  • Q1: Add a limitation section
  • A1: Thank you. We have added a limitation section in the revised manuscript. Specifically, we added: “The limitations of this study should be noted. First, we collected our sample from the northern and central parts of Taiwan; thus, results may not be generalizable for the whole country. However, we weighted our data using the whole-population outpatient database, improving the representativeness of our sample. Second, we conducted a cross-sectional survey, and hence longitudinal changes cannot be observed. Third, there are possible response and recall biases. This however, is a typical problem in survey studies.” We have added a limitations section. Please see page 7 line 218. There were some limitations of this study. First, we collected our sample from the northern and central parts of Taiwan; thus, results may not be generalizable for other parts of Taiwan. However, we weighted our data using the whole-population outpatient database, improving the representativeness of our sample. Compared with general participants, parents who had children with asthma were not more willing to spend more time or money. Our study not only investigated patients’ willingness to pay for COC but also analyzed the relationship between parents’ perception regarding and value attributed to continuity. Neither parents nor main caregivers think that changing physicians conflicts with care continuity. Future studies can continue to explore the causal linkage between continuity and value that parents place on continuity. Furthermore, researchers can investigate the perception of COC in physicians and compare providers’ characteristics, which may also be essential determinants for patients’ attitude toward their children’s COC.

  • Q2: The authors appears to generalize the findings but they are not in a position to do so based on non random selection. Suggested to add that the findings are limited to the sample tested
  • A2: Thank you for pointing this out. We agree the sample we used should not be over generalized. We have added in the Limitation section.

  • Q3: Extensive English grammar, style and format changes are required. Highly recommended to have the manuscript reviewed and edited for language by an expert. 
  • A3: We have the manuscript edited for language by an expert.

We have revised the manuscript based on these comments. All changes are tracked in a different color.Please see the attachment.

Thank you again for the valuable comments. We look forward to hear from you.

Reviewer 4 Report

This manuscript investigated caregivers’ attitude toward continuity of care (COC) and their willingness to maintain continuity for their children with asthma, using a purposive sample of 825 participants from six pediatric outpatient departments in different part of Taiwan. Perception and values of continuity of care were measured using seven items, validated by five experts and pretested by 30 respondents prior to the main survey. Results showed that more than 80% of caregivers in the asthma group opted that having a primary pediatrician would be important for their children’s health. Logistic regression showed that household income remained a significant determinant of willingness to maintain provider continuity. Caregivers’ free choices among health care providers may reduce willingness to spend extra effort to maintain high continuity of care (COC). This is a main challenge of asthma management in the Taiwanese healthcare system. The government should advocate and educate parents and caregivers about the importance of COC in asthma control.

The research agenda of this analysis is original, which examined the caregivers’ perception and value of COC. However, many errors and inconsistencies are found in the manuscript, which requires a substantial revision. Here are my specific comments:

  1. Page 2, line 54: This part described the history of NHI and current user fee of attending healthcare services in medical centers (US$14 without referral; US$5.6 with) and clinics (US$1.4). Although the relative difference is quite huge, an overseas audience (like me) have no idea whether these user fees are affordable for local population. It’d be good to include the national income statistics (e.g. GDP per capita or median household income) of the catchment area during the period.
  2. Page 2, line 70: Sampling and participants recruitment – The sample of this analysis was collected from 6 different sites in Taiwan. I had no idea about the location of these sites until in line 213 – the sites were located in northern and central part of Taiwan. I think it’s better to clearly describe the locations of these 6 sites in the methods, instead of adding some crucial information in the end.
  3. Page 2, line 72: Type of medical facilities (recruitment sites) – I am not familiar to the hospital system in Taiwan, but I’d speculate medical centers may provide the most advanced specialist outpatient services to more serious patients than others (regional hospitals, district hospitals and clinics). On the contrary, clinics may provide primary health care services to asthma patients with very mild and stable condition. I think the authors should give more details about each type of healthcare facility, instead of guessing/ speculating.
  4. If my assumption in (3) is correct, it is clearly that the WTP and willingness to spend extra time for the asthma group was strongly influenced by participants from district hospitals and clinics (see Table 3 and 4). Also, I am quite surprise that participants of the asthma group who were recruited from district hospitals and clinics were willing to pay more money (Table 3), but were less willing to spend more time (Table 4). How do the authors interpret this?
  5. Final sample size: It is reported that the final sample was 825 participants (line 84). If 147 of them had children with asthma, there should be 678 participants (825-147) had children without asthma (control group). However, the tables in the manuscript showed a different number of participants in the control group. I think the authors should revisit the number to ensure the correctness and consistency.
  6. Control group: What is the disease profile of children in the control group? We only know they had no asthma, but whether they had other serious health problems (e.g. cancer, heart disease, Type I DM, etc). These health conditions may affect the comparability of the two study groups.
  7. Results: Table 1 – please check the accuracy of the table. There were 5 participants reported zero WTP, but it accounted for 22.2% of the asthma group.
  8. Table 1: Please clarify the unit of currency (NTD/ USD).
  9. Table 1: Typo error – should be <30 minutes
  10. Table 1 reported the main results of the analysis, it should be relocated after the comparison of sample characteristics (Table 2)
  11. Table 3 and 4: please report the reference group for sub-categories under “Parents/ caregivers” and “Children” (e.g. Male sex [ref=female]; Age>40 year [ref= 18-39 years]…)
  12. Lines 164 – 178: The authors compared the odds ratios between the asthma and control group. I am wondering if the comparison is relevant or not. Epidemiological studies compare odds ratios within a regression model. Here, the odds ratio of each group was derived from two separate logistic regressions. Please revisit the analysis.
  13. Line 218: The word ‘parents’ appears twice
  14. Finally, it is obvious that the non-strict referral system under NHI is the main barrier to achieve continuity of care. Why didn’t the authors suggest a reform of NHI by setting up a strict referral system for all specialist and hospital care in Taiwan?

Author Response

Dear Editor and Reviewers,
Thank you for the valuable comments on our manuscript. They have indeed helped us enhance our manuscript.

  • Q1: Page 2, line 54: This part described the history of NHI and current user fee of attending healthcare services in medical centers (US$14 without referral; US$5.6 with) and clinics (US$1.4). Although the relative difference is quite huge, an overseas audience (like me) have no idea whether these user fees are affordable for local population. It’d be good to include the national income statistics (e.g. GDP per capita or median household income) of the catchment area during the period.
  • A1: Thank you. This information has been included in the revised manuscript. Specifically, annual per capita disposable income was approximately US$11,300 in 2018.

  • Q2: Page 2, line 70: Sampling and participants recruitment – The sample of this analysis was collected from 6 different sites in Taiwan. I had no idea about the location of these sites until in line 213 – the sites were located in northern and central part of Taiwan. I think it’s better to clearly describe the locations of these 6 sites in the methods, instead of adding some crucial information in the end.
  • A2: We have clearly described the locations of these 6 sites in the methods. Please see Page 2, line 70.

  • Q3: Page 2, line 72: Type of medical facilities (recruitment sites) – I am not familiar to the hospital system in Taiwan, but I’d speculate medical centers may provide the most advanced specialist outpatient services to more serious patients than others (regional hospitals, district hospitals and clinics). On the contrary, clinics may provide primary health care services to asthma patients with very mild and stable condition. I think the authors should give more details about each type of healthcare facility, instead of guessing/ speculating.
  • A3: We have given more details about each type of healthcare facility. Please see Page 2, line 72-75. We conducted a cross-sectional survey by using a purposive sampling design from August 2017 to February 2018. Six medical sites from the northern and central parts of Taiwan were included: two medical centers, two regional hospitals, one district hospital, and one clinic. Medical centers may provide the most advanced specialist outpatient services to more serious patients than regional hospitals, district hospitals, and clinics. Clinics may provide primary health care services to asthma patients with mild and stable condition. The six sites were chosen to represent different levels of medical institutions with a reasonably diverse geographic distribution.

  • Q4: If my assumption in (3) is correct, it is clearly that the WTP and willingness to spend extra time for the asthma group was strongly influenced by participants from district hospitals and clinics (see Table 3 and 4). Also, I am quite surprise that participants of the asthma group who were recruited from district hospitals and clinics were willing to pay more money (Table 3), but were less willing to spend more time (Table 4). How do the authors interpret this?
  • A4: Thank you for pointing this out. We think this has to do with the health system in Taiwan. In Taiwan, while medical centers provide more sophisticated medical procedures and a wider range of services, however, it is also more time consuming to go to medical centers for treatments. Clinics and district hospitals on the other hand provide reasonable quality care without complicated administrative procedures. Thus respondents recruited from clinics and district hospitals thus may involve parents/main caregivers who had higher opportunity costs for spending time. We added this interpretation in the Discussion.

  • Q5: Final sample size: It is reported that the final sample was 825 participants (line 84). If 147 of them had children with asthma, there should be 678 participants (825-147) had children without asthma (control group). However, the tables in the manuscript showed a different number of participants in the control group. I think the authors should revisit the number to ensure the correctness and consistency.
  • A5: Thank you pointing this out. The final sample size was 825, with 149 children asthma and 676 without. However, we had to exclude respondents who visited the physician for growth evaluation only because COC would not be relevant for them in this context. Thus the sample included in the regressions after coarsened exact matching was 146 for the asthma group and 547 for the non-asthma group. We have corrected the numbers and also added an explanation in the revised manuscript to avoid confusion.

  • Q6: Control group: What is the disease profile of children in the control group? We only know they had no asthma, but whether they had other serious health problems (e.g. cancer, heart disease, Type I DM, etc). These health conditions may affect the comparability of the two study groups.
  • A6: Thank you pointing this out. We did not collect information on other diseases. However, we believe this should not be a major issue because the children from the control groups are from general outpatient. In Taiwan, medical care is highly specialized, and thus if a child has a major disease, he/she is likely to be treated in specialized outpatient. However, we agree this is a possible bias in our study and we thus added this explanation in the Limitation section.

  • Q7: Results: Table 1 – please check the accuracy of the table. There were 5 participants reported zero WTP, but it accounted for 22.2% of the asthma group.
  • A7: Thank you. We have fixed the numbers in the revised version.

  • Q8: Table 1: Please clarify the unit of currency (NTD/ USD).
  • A8: We have clarify the unit of currency (NTD).

  • Q9: Table 1: Typo error – should be <30 minutes
  • A9: We have corrected the error <30 minutes.

  • Q10: Table 1 reported the main results of the analysis, it should be relocated after the comparison of sample characteristics (Table 2)
  • A10: We have relocated Table 1 and 2 and adjust the text.

  • Q11: Table 3 and 4: please report the reference group for sub-categories under “Parents/ caregivers” and “Children” (e.g. Male sex [ref=female]; Age>40 year [ref= 18-39 years]…)
  • A11: We have reported the reference group for sub-categories under “Parents/ caregivers” and “Children (Please see Table 3 and 4).

  • Q12: Lines 164 – 178: The authors compared the odds ratios between the asthma and control group. I am wondering if the comparison is relevant or not. Epidemiological studies compare odds ratios within a regression model. Here, the odds ratio of each group was derived from two separate logistic regressions. Please revisit the analysis.
  • A12: Thank you for pointing this out. We agree that it is inappropriate to compare ORs across regressions. We have modified the revised manuscript accordingly.

  • Q13: Line 218: The word ‘parents’ appears twice
  • A13: We have deleted the word ‘parents’. Please see line 218.

  • Q14: Finally, it is obvious that the non-strict referral system under NHI is the main barrier to achieve continuity of care. Why didn’t the authors suggest a reform of NHI by setting up a strict referral system for all specialist and hospital care in Taiwan?
  • A14: Thank you. This is an interesting argument. We agree that the non-strict referral system may be one of the causes of low COC in this country. However, the non-strict referral system does come with may merits (despite there are also many drawbacks). For example, as the non-strict referral system allows patients to go to any physician freely, there tends to be shorter queues and waiting time. In addition, the non-strict referral system leads to competition and leads to higher quality in certain services provided. In addition, given the current situation in the country, setting up a strict referral system is not politically feasible given the patients ate used to the current freedom for so many years. There are some debates about the necessity of provider continuity in recent studies. First, it has been argued that the government should not force a patient–physician relationship by policy. Patients must be given the freedom to choose or switch physicians if they want to; this is mainly contested by patients who do not want continuity or do not care about it. Furthermore, the chronic disease management transforms to multidisciplinary team care and it would seem like the health care system pays less attention to promoting provider continuity. There is no doubt that different groups of patients hold distinct views on the importance of continuity. That is the reason why we need to focus primarily on the parents' attitudes toward continuity in the pediatric population—a vulnerable population that has been less studied by researchers. The demand for health care among children depends on their parents/main caregivers. Although the relational continuity is a simple and effective method to improve children’s health status, there are also other factors that can contribute to patient outcome, and these factors may be changed if the current healthcare system is changed. We thus think the policy makers in Taiwan should focus on dismissing barriers that prevent continuity given the existing healthcare system in the country.

We have revised the manuscript based on these comments. All changes are tracked in a different color.Please see the attachment.

Thank you again for the valuable comments. We look forward to hear from you.

Round 2

Reviewer 1 Report

Again, no WTP value is calculated. Econometric methods exist for this.

Reviewer 4 Report

The revised manuscript has been improved and addressed most of my previous queries. Yet, the authors should improve English proficiency and accuracy, while many grammatical errors are found in the manuscript

  1. Abstract (line 19): The word “the” should be removed if the authors want to call Taiwan as a country. If the authors want to call Taiwan as a province, it should be “the Taiwan Province, ROC”
  2. Line 63: “Copayment” - it should be a small letter “c” after a comma.
  3. Line 75&78: I used “may” in my previous comment, as it was an assumption about the level of healthcare provision for these facilities. If it is the fact, the word “may” should be taken away
  4. Table 2 and Line 182-187: It is quite confusing. Which part of Table 2 was “upper” and “lower”? Line 182-187 reported the results of item 2 “Whether you have changed the pediatric physicians or not, do you think that changing physicians affect your child’s therapeutic outcomes”. It is in the middle part of Table 2, rather than the lower part. Also, will the authors want to report other figures under “Parent’s perception of continuity (unweighted) in Table 2?
  5. Line 197: “in both the groups”- the word “the” should be removed
  6. Line 270: Should be rewritten: Our study did not only investigate patients’ WTP for ….”
  7. Line 280: It is written that “the government implement the pediatric dedicated physician program” I don’t know whether it is an authors’ recommendation or it is already implemented? If it is an author’s recommendation, it should be written as “the government “should” implement…” Or used “implemented” if it is already in place
  8. Line 286: Thus, children may receive comprehensive care and ‘better’ continuity of care.